# Temporal Changes of Adsorbed Layer Thickness and Electrophoresis of Polystyrene Sulfate Latex Particles after Long Incubation of Oppositely Charged Polyelectrolytes with Different Charge Densities

**DOI:** 10.3390/polym13152394

**Published:** 2021-07-21

**Authors:** Thi Hai Yen Doan, Tien Duc Pham, Johan Hunziker, Thu Ha Hoang

**Affiliations:** 1Faculty of Chemistry, University of Science, Vietnam National University, Hanoi, 19 Le Thanh Tong, Hoan Kiem, Hanoi 100000, Vietnam; tienducpham@hus.edu.vn or; 2Graduate School of Science, Technology and Innovation, Kobe University, 1-1 Rokkodai-cho, Nada-ku, Kobe 657-8501, Japan; johan.hunziker@lion.kobe-u.ac.jp; 3University of Education, Vietnam National University, Hanoi, 144 Xuan Thuy, Cau Giay, Hanoi 100000, Vietnam

**Keywords:** long incubation, polyelectrolytes adsorption, desorption, adsorbed layer thickness, electrophoretic mobility

## Abstract

The different desorption concepts of the two polyelectrolytes PTMA5M and PTMC5M, which have similar molecular weights and differ in the charge density on the polystyrene sulfate latex (PSL) particles by 25 times, and with various charge densities in a long incubation, were systematically investigated based on hydrodynamic adsorbed layer thickness (δ_H_) and electrophoretic mobility (EPM) under two ionic strengths in the present study. Herein, in the case of highly charged polyelectrolyte PTMA5M, desorption continued for 4 h and re-adsorbing proceeded after a longer incubation time higher than 4 h. Meanwhile, in the case of lowly charged polyelectrolyte PTMC5M, an adsorption–desorption equilibrium was suggested to take into account the unchanging of both δ_H_ and EPM.

## 1. Introduction

The adsorption of polyelectrolytes on solid surfaces is commonly applied to water treatment, paper manufacturing, mineral treatment, and other industrial chemical processes [1,2]. Aggregation or stability of colloidal particles can be controlled by the polyelectrolyte adsorption induced reduction or elimination of electrostatic repulsions between particles, due to the decrement or reversal of the original surface charge of particles [3,4,5]. In addition, the adsorption of polyelectrolytes produces an increase of adsorbed layer thickness, resulting in steric repulsion, which is responsible for the stabilization of suspensions [6]. Generally, the adsorption of polycations on an anionic surface is enhanced by the high charge density of absorbate [7], the high molecular weight of absorbate [8,9] and lower salt concentrations [10], while it is limited by some factors such as steric constraints, smaller absorbent surface charge density, or a smaller surface potential [4,10].

So far, the conformation of the adsorbed polymer layer at the interface predominantly affects the flocculation or stabilization of colloidal suspensions [11,12,13,14,15]. The popular conformations of adsorbed polymer on the oppositely charged surface include loops, trains and tails [16,17,18]. The hydrodynamic adsorbed layer thickness is sensitive to the configuration of the adsorbed polymer molecules. In particular, the adsorbed layer thickness can be completely determined by the tail-conformation of the adsorbed polymer [4,19]. The polyelectrolytes can be initially presented as a loop- and/or tail-conformation at the solid–liquid interface and subsequently adopt to flatter conformations [20]. The charge density of a polyelectrolyte can modulate its conformations on oppositely charged surfaces. Hagay Kohay et al. [21] demonstrated that high-charge-density polycations prefer to directly anchor to a negatively charged montmorillonite surface in higher number of trains while low-charge-density ones preferably attach in more loops and tails. The reduction of charge density of positively charged polyelectrolyte induces that the adsorbed layer configuration on the adsorbent surface is more expanded and thicker [22,23]. The adsorbed layer thickness of low-charge-density polyelectrolytes remarkably increases with the type of surface or the higher ionic strength [10]. As a result, charged neutralization flocculation was dominant with adsorbing high-charge-density polyelectrolytes, while bridging flocculation was mainly enhanced with the adsorption of low-charge-density ones [17]. Moreover, electrophoretic mobility measurement distinguishes not only the information on the structure of the polyelectrolyte-adsorbed layer on the surface but also the roles of the electrostatic and non-electrostatic interactions in polyelectrolyte adsorption [24,25]. However, this technique has disadvantaged in order to analyze the internal structure of thick adsorbed layers [25]. Thus, an adequate complementary analytical method, such as particle tracking of Brownian motion under an optical microscope, can demonstrate adsorption kinetics of polyelectrolytes on the surface. Despite the numerous experimental and theoretical research on the effective factors underlying the adsorption and desorption polyelectrolytes with various charge density [14,22,26], the temporal behaviors of those, as well as the contributions of the electrostatic and the non-electrostatic forces, have been still unclear.

In previous research, it was found that, with a short incubation of one hour, adsorption of the high-charge-density polyelectrolytes with higher molecular weights in the lower ionic strength was kinetically controlled but not in the higher ionic strength, regardless of the polyelectrolyte molecular weight [9,26]. The later phenomena under a high ionic strength condition have yet to be quantified. In addition, our previously published study [27] indicated that a more stable adsorbed layer of high-charge-density polyelectrolyte, which was formed closely to the core particle surface, could not be removed out of the surface, while the adsorbed layer of the low-charge-density polyelectrolyte easily removed via the perpendicular direction to the polystyrene sulfate latex (PSL) particle surface in a short incubation of one hour under the overshooting condition. Another key point that was inferred was that the stable layer immediately decided on the electrophoretic mobility (EPM). However, the mechanisms underlying different charge density polyelectrolytes on the particle surface after a long incubation time have also not been explained.

In the present study, adsorption dynamics were comprehensively solved for both highly- and lowly-charged polycations on oppositely charged PSL colloidal particles after a long incubation of 1 week. The hydrodynamic adsorbed layer thickness and electrophoretic mobility of the PSL particles with/without adsorbing polyelectrolytes were carried out as a function of time lapse under different ionic strength conditions. Based on the observed relationship between the adsorbed layer thickness and the electrophoretic mobility, different behaviors of polyelectrolytes differing of the charge density on the PSL colloidal particles under long incubation are also proposed.

## 2. Materials and Methods

### 2.1. Materials

In this study, polystyrene sulfate latex particles (PSL, Thermo Fisher Scientific, Inc., Waltham, MA, USA) with a diameter of about 1.2 ± 0.01 μm and a surface charge density of 5.5 μC·cm^−2^ were employed as negatively charged colloidal particles. Before each adsorption experiment, sonication of the PSL colloidal particles was conducted for 25 min to eliminate particle aggregations. The PSL colloidal particles with initial concentration of 10^8^ particles·cm^−3^ and initial concentration of particle sulfate groups of 1.8 × 10^14^ cm^−3^ were applied in all experiments. Two linear polycations with similar molecular weight and a charge density difference of 25 times were used as absorbates in adsorption measurements. The polyelectrolytes are poly((2dimethylamino) ethyl methacrylate) methyl chloride quaternary salt (PTMA5M; Kaya Floc Co. Ltd., Tokyo, Japan) and a co-polymer of PTMA5M and acrylamide (PTMC5M; Kaya Floc Co. Ltd.) corresponding to similar nominal molecular weights of 4.9 × 10^6^ and 5.2 × 10^6^ g·mol^−1^. The charge densities, ρ of PTMA5M and PTMC5M were 100% and 4%, respectively. The chemical structures of two polycations were described in [27]. Stock polyelectrolyte solutions of 100 mg·L^−1^ were prepared by stirring an appropriate amount of polycations with potassium chloride solutions for 72 h. Working solutions diluted from the stock solutions were used for a week. All polyelectrolyte solutions were kept at approximately 5 °C in darkness to eliminate the light exposure-induced degradation. Potassium chlorine solutions at the concentration of 0.1 and 10 mM were applied to maintain the ionic strength. Before conducting the experiments, the electrolyte solutions were filtered by using a 0.2 μm cellulose membrane. Ultrapure water produced from Elix Advantage 5 (Millipore) with electric conductivity around 0.6 µS·cm^−1^ was employed in all solutions and measurements.

### 2.2. Methods

Determination of hydrodynamic layer thickness was by mean of particle tracking of Brownian motion.

The Brownian movements of single PSL colloidal particles with/without adsorption of polyelectrolytes in one dimension were tracked by using a CCD video camera (WV-BL200, Matsushita Co. Ltd., Yamanashi, Japan) equipped with an optical microscope (BX50, Olympus). The method was described in more detail in [26,27]. The diffusion coefficient of single PSL particle *D* was determined using the Einstein relation [26,28]:(1)$$\bar{\Delta x^{2}}=2D\tau$$
where $\bar{\Delta x}$ (m) is the particle displacement calculated from the center of mass at time interval of *τ* (s).

Then hydrodynamic layer thickness of adsorbed polyelectrolytes was calculated by comparing *D* of the bare single PSL particle with that of a particle with adsorbing polyelectrolyte using the Stokes–Einstein equation, as shown in the following Equations (2) and (3) below, respectively.
(2)$$D=\frac{k_{B}T}{6\pi \alpha \mu }$$
(3)$$D=\frac{k_{B}T}{6\pi \left(\alpha +\delta H\right)\mu }$$
where *k_B_* is Boltzmann constant (*k_B_* = 1.38 × 10^−23^ J·K^−1^), T is the absolute temperature (K), *α* is hydrodynamic radius (m), *μ* is the solvent viscosity (Pa.s) and *δ_H_* (m) is the hydrodynamic layer thickness of adsorbed polyelectrolytes.

### 2.3. Experimental Procedure

A polycation volume of 5.0 mL under two ionic strength conditions was added to one side of a forked flask while in the other side, the same volume of the initial PSL concentration of 10^8^ particles cm^−3^ was introduced. An end-over-end apparatus described in the previous reference [9] was used to mix the samples for 10 times at a frequency of 1 Hz (Figure 1). The experiment procedure was mentioned in more detail in [27]. Lapse of time was calculated from stopping mixing the samples. The experimental measurements were carried out at room temperature (293 ± 2 K), controlled with an air conditioner.

#### 2.3.1. Experimental Measurements of Hydrodynamic Layer Thickness of Adsorbed Polyelectrolytes

After mixing and fixed incubation time, the samples were added to a 0.2 mm width × 2.0 mm length rectangle capillary tube. The Brownian movements of a single colloidal PSL particle were surveyed by using an optical microscope with a magnitude of 80× and a time resolution of 0.1 s. The video of Brownian movements of a colloidal particle was recorded with 500 frames and 1000 displacements for about 0.033 s. The clear images of the PSL particle were maintained by the manual microscope focus.

#### 2.3.2. Electrophoretic Mobility Measurements

The electrophoretic mobility of PSL particles (EPM) (µm·cm·V^−1^·s^−1^) was measured in an electric field of 11.3 (V·cm^−1^) by the Zeta-sizer Nano-ZS setup, with a laser velocimetry (Malvern Instruments Ltd., Worcestershire, UK). The samples were introduced into a plastic capillary cell and then inserted in the analyzer. Each sample was measured for at least 3 times. Each measurement was carried out with 11 sub-runs.

#### 2.3.3. Viscosity Measurements

The viscosity of the solutions was determined by a Cannon–Fenske capillary viscometer (S0-94331, Model 100, SIBATA Ltd., Saitama, Japan). The viscosity experiments were conducted at 293K using a controlled temperature bath. The sample was added into a viscometer, inserted to the bath, and stood for 5 min to get a temperature equilibration before starting each experimental measurement. Relative viscosity, *η_r_* was determined by comparing the time of polymer solution, *t_s_*, and the time of solvent, *t*_0_, flowing through the viscometer.
(4)$$\eta_{r}=\frac{t_{s}}{t_{0}}$$

At low polyelectrolyte concentration, polyelectrolyte existed as a random coil in the solution and assumed as a considerable sphere. The charge density of electrolyte ions was 1000 mmol·L^−1^ at the KCl concentration of 1 M, while the charge densities of PTMA5M and PTMC5M calculated were approximately 569.02 and 13.41 mmol·L^−1^, respectively, at 5 ppm of polyelectrolyte concentration. Therefore, the polyelectrolyte charges were compensated by oppositely charged electrolyte ions, considering as a neutral particle in the solution. The size of the polyelectrolyte was calculated by Einstein’s viscosity equation, applied for neutral spherical particles [29]. The Einstein’s equation is given as the following (5):(5)$$\eta_{r}=\frac{\eta_{s}}{\eta_{0}}=1+\frac{5}{2}\Phi$$
where *η_s_* is the viscosity of polymer solution (N·s·m^−2^), *η*_0_ is the solvent viscosity (N·s·m^−2^), *Φ* is the volume fraction.

Then, the hydrodynamic diameter of polymer was calculated by:(6)$$d=10^{9}\times {\left[\frac{\left(\eta_{r}-1\right)\times M_{w}}{\frac{5}{2}\times \left(\frac{\pi }{6}\right)\times C\times N_{A}})\right]}^{\frac{1}{3}}$$
where *d* is the hydrodynamic diameter of polymer in solution (nm), *M_w_* is the molecular weight of polymer (g·mol^−1^), *C* is the polymer concentration (ppm), *N_A_* is the Avogadro number (*N_A_*= 6.02 × 10^23^ mol^−1^).

## 3. Results and Discussion

### 3.1. Contributions of Electrostatic and Non-Electrostatic Interactions on Adsorption of Polyelectrolytes with Different Charge Densities onto PSL Particles at Different Ionic Strengths

Adsorption of polyelectrolytes with high- and low-charge densities onto the positively charged PSL particles immediately occurred when polycations were added to the colloidal suspension. The primary driving forces for the adsorptions of PTMA5M and PTMC5M are mainly electrostatic attractions between particle surface and oppositely charged polyelectrolyte segments [17,30,31] and non-electrostatic interactions such as hydrogen bonding, hydrophobic interaction, and ion bindings [24,32,33]. Furthermore, the non-electrostatic interactions between polyelectrolyte segments can be attributed to an increase layer thickness, which continues after reaching an isoelectric point (IEP) [24]. It can be inferred that the hydrodynamic diameter of the particle increases with the decreasing absolute electrophoretic mobility of particles, and this even leads to a reversal of particle charge when adsorbing an excess dosage of polycations (Figure 2). This phenomenon is also clearly affirmed in the previous research [27]. As a result, the enhancement of aggregation, or the stability of the colloidal suspension, occurs after adding the positively charged polyelectrolytes to the oppositely charged PSL particles. The increments in hydrodynamic particle diameter due to adsorption of polyelectrolytes were evaluated by calculating the difference between the diffusion constant of a bare PSL particle and another single PSL particle with the adsorbed polycations.

Figure 2 shows that, at a fixed ionic strength, the adsorbed layer thickness as well as the EPM after 5 min of incubation dramatically increased when increasing the initial polyelectrolyte concentration from 1 to 10 ppm. This increment could be due to kinetically-controlled adsorption and/or either a dynamic adsorption–desorption equilibrium or a partial surface coverage [19,34,35]. At the beginning, a higher concentration of polyelectrolytes in the solution was more numerous as polyelectrolytes diffused from the solution to the PSL particle surface [19,35]. This led to polyelectrolyte attachments to the particle surface, consisting of adsorbed polyelectrolyte chains with new conformations [32], and further resulting in an increment of adsorbed layer thickness. On the other hand, the value of IEP extrapolated from Figure 2 was approximately 0.28 and 0.29 ppm in the case of the high-charge-density polyelectrolyte PTMA5M, while the value was about 0.6 and 0.8 ppm in the case of low-charge-density polyelectrolyte PTMC5M, at high and low ionic strengths, respectively. In the range of the initial polyelectrolyte concentration of 2 to 10 ppm, the total adsorbed polyelectrolyte charges over-compensated the charges of the particle surface. Thus, the explanation of a polyelectrolyte partially-coated particle surface was excluded. Moreover, the IEP value of the highly charged polyelectrolyte was lower than the lowly charged one, due to a faster adsorption process of PTMA5M, resulting in more than a 25-fold higher charge density than PTMC5M. These results are in agreement with the almost previous findings [22,23].

Considering the polyelectrolytes adsorption on an oppositely charged surface driven by the electrostatic interactions, the adsorption regimes [36] were limited by the shielding that induced by not only electrostatic repulsions between the positively charged groups in polyelectrolyte chains but also in the electrostatic polyelectrolyte segment-PSL surface attractions when the ionic strength increased. In the former effect, polyelectrolytes shrunk in correlation with a reduction of relative viscosity when the KCl concentration increased from 0.1 mM to 1 M (Table 1). It should be noted that polyelectrolytes are less coiled in solution, inducing less swollen adsorbed layer. However, this effect was negligible for the low-charge-density polyelectrolyte because of a significant distance between the charged groups on the polyelectrolyte chain (approximately 25 nm). Therefore, the relative viscosity of PTMC5M measured as well as the hydrodynamic thickness of adsorbed layer had not much difference between all ionic strengths. In the lateral effect, the attractive interactions, not only between the polyelectrolyte segments and the PSL particles, but also between polyelectrolyte chains, were all reduced, corresponding to a reduction of δ_H_ as well as the PSL surface charge, as evidenced by a decrease in the EPM. The tendency of the attractive forces to ionic strength was similar to polyelectrolyte adsorption on the surface with opposite charge [37]. These results suggest that the electrostatic attraction is the exclusive driving force for the adsorption of the highly positively charged PTMA5M onto the oppositely charged PSL surface.

However, contrary to the tendency of the highly charged polyelectrolyte PTMA5M, it can be observed in Figure 2 that there was an increase in the adsorbed layer thickness of the lowly charged polyelectrolyte PTMC5M, increasing the ionic strength. PTMC5M even adsorbed continuously after reaching the IEP. It was concluded that there are additional non-electrostatic interactions, not only between the PSL particle surface and the polyelectrolyte segments, but also between polyelectrolyte chains [37,38]. The non-electrostatic forces were very important for the adsorption process of low-charge-density polyelectrolytes [33]. Some authors also found that the thicker adsorbed layer was plotted as a function of increment of the Debye–Hückel parameter, κ^−1^ [36,39,40]. It could be suggested that not only were the hydrophobic interactions between surface and polyelectrolytes promoted, but the repulsive forces between polyelectrolyte segments were also reduced when the ionic strength increased. This resulted in the main increment of the adsorbed layer thickness, as well as the more and/or longer extended polyelectrolyte conformation of protruding loops and tails into the solution [4,38]. Therefore, the non-electrostatic force plays a significantly important role in the PTMC5M adsorption, resulting simultaneously in attaching loosely to the PSL particle surface and strongly to other PTMC5M, as well as exhibiting a swelling conformation when the ionic strength increases. Therefore, the adsorbed amount of the low-charge-density polyelectrolyte was reduced upon addition of salt in almost of cases [10].

The sign of PSL electrophoretic mobility changed from negative to positive, passing through the IEP. These results from the adsorption of the oppositely charged polyelectrolytes are illustrated in Figure 2. A shift in the IEP can be observed as the concentration of KCl increased 100 times from 0.1 to 10 mM. The IEP value shifted to a higher value as the ionic strength increased, proving the limited adsorption of PTMA5M. Oppositely, the lower IEP value at the higher ionic strength confirms that more PTMC5M adsorbed onto the PSL colloidal suspension due to promotion of hydrophobic interactions under the high ionic strength condition.

### 3.2. Relaxation Behaviors of Adsorbed Layer of Polyelectrolytes with Different Charge Density on the PSL Particles in a Long Incubation

In our previous research [27], the non-equilibrium states of various charge density polycations on the negatively charged PSL colloidal particles were clarified. It was confirmed that the δ_H_ of PTMA5M on the PSL particle surfaces decreased while EPM of PTMA5M-adsorbed PSL particles kept constant under the overshooting condition in a short incubation of 1 h. Meanwhile, the δ_H_ of PTMC5M maintained while the EPM decreased slightly with an excess of PTMC5M dosage in the time scale of 1 h. Moreover, the different concepts corresponding with the desorption of the different-charge-density polyelectrolytes were proposed. That is, the high-charge-density polyelectrolyte PTMA5M chains were desorbed sequentially from the outer layer to the inner layer (which is closer to the PSL particle core), while the low-charge-density polyelectrolyte PTMC5M chains perpendicularly aligned to the PSL surface were removed. In the present study, the behaviors of the positively charged polyelectrolytes with various charge densities on the PSL particle surface in a long incubation from 1 h to 1 week were thoroughly investigated.

At the given polyelectrolyte concentration, the δ_H_ and the EPM values were changed against the time lapse in the time scale of 1 week at both ionic strengths. There are herein some different tendencies. In the case of PTMA5M adsorption, the δ_H_ dramatically decreased from 1 to 4 h of the time lapse in two ionic strengths, while the value of EPM changed insignificantly (Figure 3). Re-conformation and/or detachment of polyelectrolytes might explain the decrement of the adsorbed layer thickness [20,41]. The EPM clearly suggests that the concepts could be responsible for this decrement of the adsorbed layer thickness. According to Ohshima and Kondo’s theory, the electrophoretic mobility could depend on total charge amount contained inside the whole adsorbed layer, the ionic shielding effect of electrolytes, the adsorbed layer thickness (δ_H_), and a uniform density of fixed-charge groups (ρ_fix_) distributed inside the adsorbed layer [42,43,44,45,46,47]. If the adsorbed polyelectrolytes were re-conformed on the PSL particle surface, the total amount of charges remained constant, while the area in which fewer electrolyte ions penetrated was restricted, as the adsorbed thickness changed into a flatter-conformation, decreasing the electrolyte ion shielding effect [46]. As a result, the electrophoretic mobility could increase.

Herein, the EPM did not change significantly in the range of time lapse from 1 to 4 h, suggesting a desorption process. Moreover, there was a significant increment of the δ_H_ after 4 h of incubation, potentially due to re-adsorption, as the remaining PTMA5M molecules were still not eliminated from the solution. Meanwhile, the EPM remained constant at two ionic strengths. This tendency of the EPM after 4 h incubation could be explained by the fixed charge group density distributed inside adsorbed layer and/or the electrolyte ion shielding effect. As immediately mentioned above, the more penetrative the electrolyte ions were during the increment of the δ_H_, the stronger the shielding effect of electrolyte ions was. As a result, the value of the EPM ought to decrease. However, the EPM remained unchanged against the significant changes of the δ_H_. It was inferred that the shielding effect of the electrolyte ions was negligible, hence the EPM was totally dependent on the ρ_fix_ and remained. Summarily, desorption and re-adsorption takes account for the decrement and the increment of the δ_H_ in a short incubation lower than 4 h and a long incubation higher than 4 h, respectively, while the fixed charge groups inside the PTMA5M-adsorbed layer resulted in a constant EPM value.

In addition, the mobility trend could be explained on the basis of the dependence of the electrophoretic mobility of soft particles on the ratio of adsorbed layer thickness and the Debye length (δ_H_.κ) [42,45]. When the ratio (δ_H_.κ) was higher than unity at both ionic strengths (δ_H_.κ >> 1), the EPM mainly depended on the uniform density of fixed-charge groups within the polyelectrolyte layer, and was insensitive to the adsorbed layer thickness. In the case of PTMC5M adsorption, both the adsorbed layer and the EPM remained constant in a long incubation of 1 week under two ionic strength conditions (Figure 4). One more time, desorption could affirm to be a reason. The low-charge-density polyelectrolytes easily detached perpendicularly from the PSL surface, introducing the constant adsorbed layer thickness and the decrement of the EPM in the short incubation of 1 h, observed in our previous study [27]. However, herein, the EPM remained constant in the longer incubation. After the long incubation, it can be suggested that an adsorption–desorption equilibrium was reached. As a result, a fixed value of the δ_H_ (about 100 nm) was obtained while the fixed-charge groups inside the adsorbed layer were responsible for the uncharged value of the EPM.

### 3.3. Adsorption/Desorption Concepts of Polyelectrolytes with Different Charge Density on the PSL Particles in a Long Incubation

Two previously proposed concepts for the desorption of two polyelectrolytes with different charge densities on the PSL colloidal particles in the short incubation of 1 h [27] were affirmed by the data found in the present study. However, the adsorption behaviors of the different charge density polycations on the oppositely charged PSL particles with a long incubation of 1 week were systematically investigated, based on the changes of the hydrodynamic adsorbed layer thickness and the EPM. Formerly, the detachment of the homogenous adsorbed layer of PTMA5M from the outer to the inner took into account the simple reduction of thickness, while the fixed-charge groups responded to the remaining EPM of PTMA5M adsorbed-PSL particles through 4 h under two ionic strength conditions. When the adsorption was mainly driven by the electrostatic interactions, i.e., PTMA5M, the inner-adsorbed layer, which is close to the PSL surface, tightly attached, while the outer-adsorbed layer of loose attachments to the particle surface easily desorbed. This results from the weaker non-electrostatic interaction between polyelectrolyte segments, rather than the electrostatic ones between particles and polyelectrolytes. The desorption process continued happening until reaching a fixed δ_H_ (about 20 nm), suggesting the presence of a thin, stiff adsorbed layer. A long desorption process of PTMA5M (about 4 h) was suggested. Moreover, the re-adsorption occurred after 4 h of incubation, as the free PTMA5M was still not removed from the solution, inducing the increment of the δ_H_. The mobility was constant in the long incubation of 1 week due to the ρ_fix_. The schematic behaviors of the high-charge-density PTMA5M on the PSL colloidal particles with the negative charges are illustrated in Figure 5.

Latterly, in the PTMC5M adsorption of the short incubation of 1 h, the removal of alternative parts of the adsorbed polyelectrolyte from the PSL surface introduced the decrement of the average charge density and a constant thickness. Herein, some parts of PTMC5M weakly made contact with the PSL surface due to hydrophobic interactions—desorption occurred easily. In the long incubation of more than 1 h, it was suggested that an adsorption–desorption equilibrium was reached in the case of PTMC5M adsorption. Then, the fixed values of both the adsorbed layer thickness and the EPM were reached. The schematic behaviors of polycation PTMC5M on the PSL surface are illustrated in Figure 6.

## 4. Conclusions

The adsorption behaviors of polycations, with 25-fold differences in charge density and with similar molecular weights, onto oppositely charged PSL particle surfaces were comprehensively investigated in this study. This included the temporal changes of the hydrodynamic-adsorbed layer thickness and the electrophoretic mobility under the different ionic strength conditions, with a long incubation of 1 week. In the case of the adsorption of the highly charged polyelectrolyte PTMA5M, fast adsorption occurred at initial times, followed by desorption for about 4 h, and finally a re-adsorption process was observed. On the other hand, an adsorption–desorption equilibrium was suggested in the case of the lowly charged polyelectrolyte PTMAC5M. It resulted in a fixed value of the adsorbed layer and a constant mobility. The value of electrophoretic mobility followed Ohshima’s theory and was dependent on the fixed charge groups within the adsorbed layer and/or the electrolyte ion effect.

## Figures and Tables

**Figure 1 polymers-13-02394-f001:**
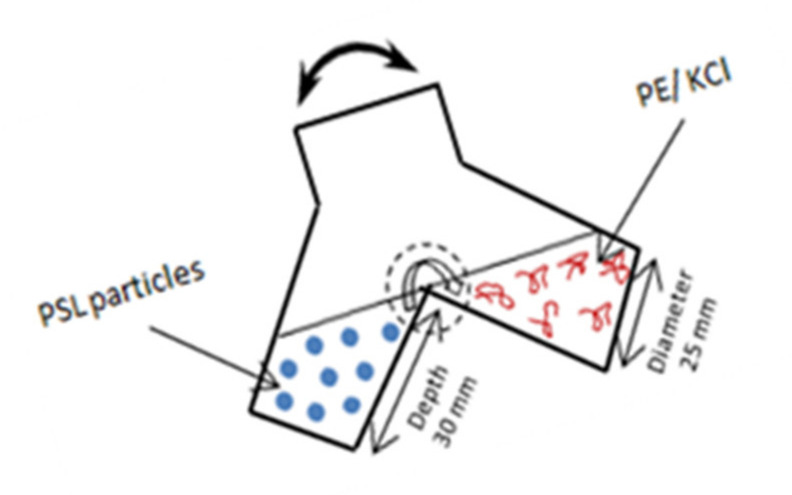
Schematic illustration of an end-over-end apparatus used for mixing samples.

**Figure 2 polymers-13-02394-f002:**
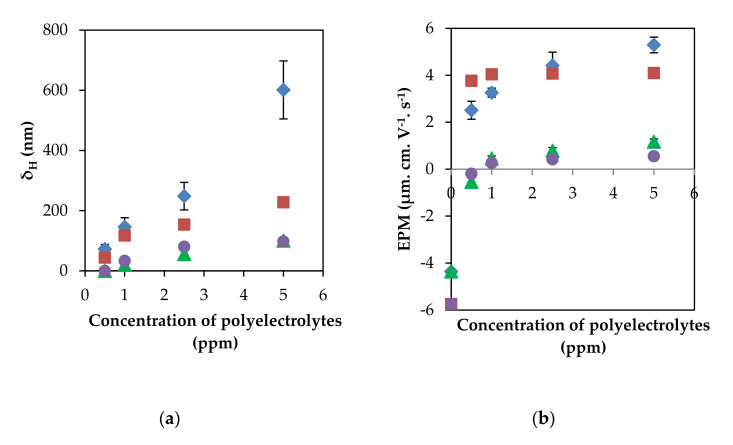
Variation of: (**a**) the hydrodynamic-adsorbed layer thickness, δ_H_ and (**b**) the electrophoretic mobility of polyelectrolyte-adsorbed PSL particles, EPM acting as a function of the polyelectrolyte dosage after 5 min of incubation at different salt concentrations, including 0.1 mM KCl: PTMA5M (♦), PTMC5M (▲) and 10 mM KCl: PTMA5M (■), PTMC5M (**●**). The error bars denoted the standard deviations of at least three different runs.

**Figure 3 polymers-13-02394-f003:**
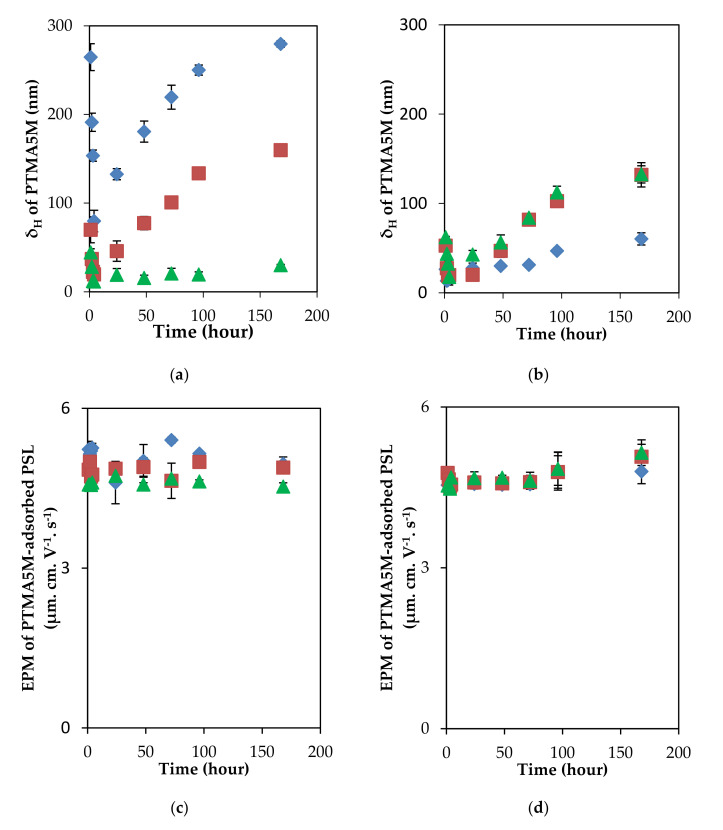
δ_H_ and EPM of PTMA5M–adsorbed PSL colloidal particles as a function of time lapse under two ionic strength conditions. In 0.1 mM KCl: (**a**) δ_H_, (**c**) EPM and in 10 mM KCl: (**b**) δ_H_, (**d**) EPM. Initial PTMA5M concentrations: 10 ppm (♦), 5 ppm (■), 2 ppm (▲). The deviations of different runs are shown by the error bars.

**Figure 4 polymers-13-02394-f004:**
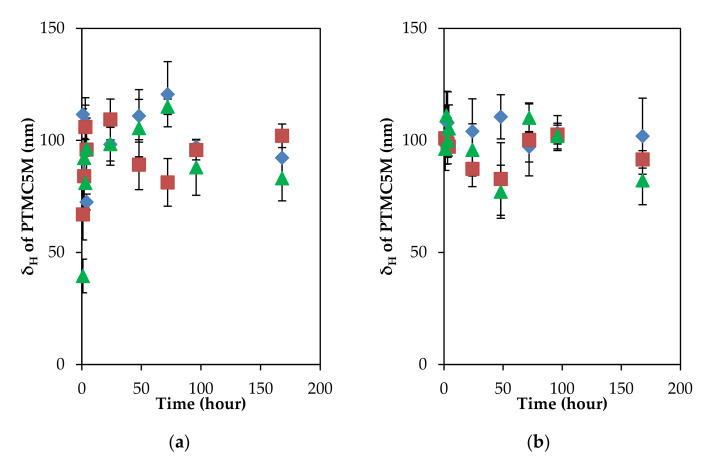
δ_H_ and EPM of PTMC5M–adsorbed PSL colloidal particles as a function of time lapse under two ionic strength conditions. In 0.1 mM KCl: (**a**) δ_H_, (**c**) EPM and in 10 mM KCl: (**b**) δ_H_, (**d**) EPM. Initial PTMC5M concentrations: 10 ppm (♦), 5 ppm (■), 2 ppm (▲). The deviations of different runs are shown by the error bars show.

**Figure 5 polymers-13-02394-f005:**
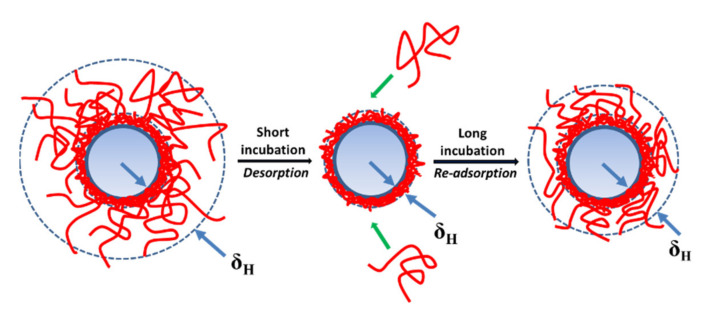
The polycation PTMA5M behaviors on the PSL colloidal particles with opposite charge in the long incubation.

**Figure 6 polymers-13-02394-f006:**
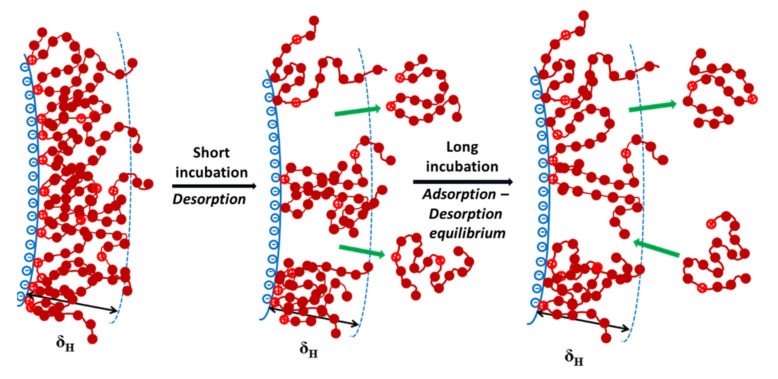
The polycation PTMC5M behaviors on the PSL colloidal particles with opposite charge in the long incubation.

**Table 1 polymers-13-02394-t001:** Relative viscosity and hydrodynamic diameter of PTMA5M and PTMC5M at 5 ppm determined by Einstein’s equation at different KCl concentrations [27]. The standard deviations were calculated by at least eight experiments.

Polyelectrolytes	Relative Viscosity			Hydrodynamic Diameter (nm)
	0.1 mM KCl	10 mM KCl	1 M KCl	1 M KCl
PTMA5M, ρ = 100%	1.0260 ± 0.0012	1.0089 ± 0.0009	1.0030 ± 0.0005	155.46 ± 8.97
PTMC5M, ρ = 4%	1.0063 ± 0.0008	1.0062 ± 0.0007	1.0038 ± 0.0004	171.91 ± 6.69

## Data Availability

Not applicable.

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
