# Peer review of "Temporal Changes of Adsorbed Layer Thickness and Electrophoresis of Polystyrene Sulfate Latex Particles after Long Incubation of Oppositely Charged Polyelectrolytes with Different Charge Densities"

_polymers, 2021, doi:10.3390/polym13152394_

Round 1

Reviewer 1 Report

In this manuscript, the authors explore the important effect of charge density on the desorption-adsorption behavior of polyelectrolytes. This study is a nice one that will have a great contribution to both fundamental research and practical application. Also, the manuscript is well-written, with solid supports from the authors’ experimental results. I feel that this manuscript deserves publication after addressing the following questions.

  1. When mentioning high and low charge density, the authors need to calculate the detailed value of charge density. The value can be used as a “standard” for other researchers, to compare with other polyelectrolyte systems.
  2. In table 1, the authors need to add error bars for the Rh. Also, I think the accuracy is not correct, for example, 318.64 means the accuracy of ±0.01 nm. No instrument can give such accurate number.
  3. The authors mention non-electrostatic, what types of special non-electrostatic? Hydrophobic interaction? Van der Waals force? Need clarification.
  4. In Figure 2b, there is a switch from negative charge to positive charge. That means the nanoparticles continuously absorb more and more polyelectrolytes, and leads to charge reverse? Can the authors provide some explanations? Or include some refs? This may help readers to understand the phenomena.
  5. The first sentence of introduction is difficult to understand. Please make it concise.
  6. Although not necessary, it is better to use more nanoparticles with the charge density between 4% and 100%. Two sample points seem to be not very strong.

Author Response

Dear reviewers

Thank you so much for your effective comments to improve our manuscript. We carefully examined the raised comments, the manuscript was accordingly revised.

All revisions are marked in red.

Reviewer 2 Report

The manuscript studies the adsorption of two polyelectrolytes with different charge densities onto the surface of oppositely charged latex particles. Apparently, it is a continuation of the study published in ref. 27 and, although it is not original, the text is mostly clear and concise, making it publishable. Hence, I support its publication with some revisions, as stated below.

1. What is the surface charge density, i.e. the charge concentration, for the latex particles? The supplier might inform the concentration of sulfate groups, so this should be included in the new version of the manuscript.

2. Information on dispersity (or polydispersity) of the polyelectrolytes should also be included.

3. Does the model used to estimate Einstein's viscosity and polymer diameter consider that the particles are charged? I suppose it considers that the particles are neutral and some correction factor should be considered. Please, specify that.

4. The polyelectrolyte stiffness should also be considered in the discussion of the results. How could this parameter affect the observed phenomena for each polyelectrolyte?

5. Caption of Figure 2 states a "temporal variation" but actually no parameter related to time is evaluated. Please, correct that. In the same figure, the incubation time should be expressed.

6. In Figure 3 and 4, data for polyelectrolyte concentration of 0 ppm are not presented. I suppose it does exist, since it would imply a solution with no polyelectrolyte. Please, correct that.

7. Figures 1, 5 and 6 also appear in ref. 27, so it should be mentioned that they are adapted from this reference.

8. Some polishing in english text is recommended. 

Author Response

(The authors gave the same response as above.)
